# Phenotyping Indices of CYP450 and P-Glycoprotein in Human Volunteers and in Patients Treated with Painkillers or Psychotropic Drugs

**DOI:** 10.3390/pharmaceutics15030979

**Published:** 2023-03-18

**Authors:** Léa Darnaud, Clément Delage, Youssef Daali, Anne-Priscille Trouvin, Serge Perrot, Nihel Khoudour, Nadia Merise, Laurence Labat, Bruno Etain, Frank Bellivier, Célia Lloret-Linares, Vanessa Bloch, Emmanuel Curis, Xavier Declèves

**Affiliations:** 1Biologie du Médicament—Toxicologie, AP-HP, Hôpital Cochin, 27 rue du Faubourg St. Jacques, 75679 Paris, France; 2Faculty of Health, Université Paris Cité, Inserm, UMRS-1144, Optimisation Thérapeutique en Neuropsychopharmacologie, 75006 Paris, France; 3Service de Pharmacie, Hôpital Lariboisière—Fernand Widal, AP-HP, 75010 Paris, France; 4Division of Clinical Pharmacology and Toxicology, Department of Anesthesiology, Pharmacology, Intensive Care and Emergency Medicine, Geneva University Hospitals, 1205 Geneva, Switzerland; 5Centre de la Douleur, AP-HP, Hôpital Cochin, 75679 Paris, France; 6Laboratoire de Toxicologie, Hôpital Lariboisière, AP-HP, 75010 Paris, France; 7Département de Psychiatrie et de Médecine Addictologique, Hôpital GHU Lariboisière-Fernand Widal, AP-HP, 75010 Paris, France; 8Ramsay Santé—Hôpital Privé Pays de Savoie, 74100 Annemasse, France; 9Faculté de Pharmacie de Paris, Université Paris Cité, UR 7537 BioSTM, 75006 Paris, France; 10Laboratoire d’hématologie, Hôpital Lariboisière, AP-HP, 75010 Paris, France

**Keywords:** CYP450, P-glycoprotein, cocktail drugs, phenotyping, pharmacokinetics

## Abstract

Drug-metabolizing enzymes and drug transporters are key determinants of drug pharmacokinetics and response. The cocktail-based cytochrome P450 (CYP) and drug transporter phenotyping approach consists in the administration of multiple CYP or transporter-specific probe drugs to determine their activities simultaneously. Several drug cocktails have been developed over the past two decades in order to assess CYP450 activity in human subjects. However, phenotyping indices were mostly established for healthy volunteers. In this study, we first performed a literature review of 27 clinical pharmacokinetic studies using drug phenotypic cocktails in order to determine 95%,95% tolerance intervals of phenotyping indices in healthy volunteers. Then, we applied these phenotypic indices to 46 phenotypic assessments processed in patients having therapeutic issues when treated with painkillers or psychotropic drugs. Patients were given the complete phenotypic cocktail in order to explore the phenotypic activity of CYP1A2, CYP2B6, CYP2C9, CYP2C19, CYP2D6, CYP3A, and P-glycoprotein (P-gp). P-gp activity was evaluated by determining AUC_0–6h_ for plasma concentrations over time of fexofenadine, a well-known substrate of P-gp. CYP metabolic activities were assessed by measuring the CYP-specific metabolite/parent drug probe plasma concentrations, yielding single-point metabolic ratios at 2 h, 3 h, and 6 h or AUC_0–6h_ ratio after oral administration of the cocktail. The amplitude of phenotyping indices observed in our patients was much wider than those observed in the literature for healthy volunteers. Our study helps define the range of phenotyping indices with “normal” activities in human volunteers and allows classification of patients for further clinical studies regarding CYP and P-gp activities.

## 1. Introduction

Drug-metabolizing enzymes and drug transporters are key determinants of drug pharmacokinetics (PK) and subsequent drug response in humans. Among these proteins, cytochromes P450 (CYP), widely expressed in various tissues such as the gut and liver, are implicated in drug absorption and elimination. CYP metabolize more than three-fourths of small-molecular-weight drugs available on the market [1]. P-glycoprotein (P-gp), the most widely studied ATP-binding cassette (ABC) drug efflux transporter, is highly involved in the absorption, distribution, and excretion of drugs, as it is abundantly expressed in various organs such as the small intestine, tissue barriers, kidneys, and liver [2]. The high expression of CYP and P-gp in enterocytes and hepatocytes contributes to the overall intestinal and hepatic first-pass effect of drugs administered by the oral route. Metabolic activity of CYP is highly variable in humans, resulting in an important interindividual variability in drug PK and response. Variability factors include demographic parameters, genetic polymorphisms, and epigenetic variations, but they also include exposure to environmental factors such as co-medicated drugs that have inhibition or induction effects on CYP, food and beverages, and exposure to pollutants [3]. Drug influx and efflux transporters are also important sources of variability in drug PK and pharmacodynamics [4]. Even though genetic variants of CYP and P-gp contribute to the variability of drug PK and response, the genotyping approach has several limitations, as it does not take into account other sources of variability such as demographic parameters, disease-state conditions, exposure to environmental factors, and drug–drug interactions. In this regard, in vivo phenotypic evaluation of CYP and P-gp activities has the advantage of considering both genetic and non-genetic factors, and it can be easily conducted in a clinical setting. The cocktail-based CYP and transporter phenotyping approach, consisting in the administration of multiple CYP or transporter-specific probe drugs, is developed to determine their activities simultaneously in human subjects. Whereas several cocktails for CYP phenotyping have been developed over the past two decades [5,6,7,8,9,10,11,12,13,14,15,16,17], those assessing transporters [18] or both transporters and CYPs are still marginal [19,20]. Some phenotyping metabolic and transport indices were established as accurate evaluations of CYP and/or transporter activity based on “normal” CYP and P-gp function in human volunteers, in the presence of CYP and P-gp inhibitors or inducers [21]. These phenotyping indices were mostly established based on a limited number of healthy volunteers [21,22], often young, and it is less known whether or not these indices are applicable to heterogeneous populations of patients with several co-medications.

This article is divided into two parts. First, we conducted a literature review to analyze the phenotypic assessment of healthy volunteers using a cocktail approach where at least one probe drug is part of the Geneva cocktail, aiming to better define phenotypic indices for poor, normal, or extensive metabolizers. Second, we retrospectively compared the results of phenotypic assessment we performed in patients facing therapeutic issues, such as important adverse effects or lack of efficacy with painkillers or psychotropic drugs, to the phenotypic indices we obtained from the literature analysis in order to describe these indices in clinical situations.

## 2. Materials and Methods

### 2.1. Part 1: Determination of Phenotypic Indices from a Literature Review

#### 2.1.1. Studies Selection

We performed a literature review of studies assessing CYP1A2, CYP2B6, CYP2C9, CYP2C19, CYP2D6, CYP3A, and/or P-gp using drugs from the Geneva cocktail: caffeine, bupropion, flurbiprofen, omeprazole, dextromethorphan, midazolam, and/or fexofenadine. We searched the PubMed database up until April 2021, using logical combinations of the following terms: phenotyping, drug cocktails, clinical pharmacokinetics, each drug of the Geneva cocktail. Case reports were not included.

#### 2.1.2. Data Collected

The following data were collected: drugs in the cocktail, number of subjects, dose administered, plasma analytical method, phenotyping indices (arithmetic or geometric mean, standard deviation, 90% or 95% confidence intervals or median). Pharmacokinetic data (T2h, T3h, T6h, AUC_0–6h_) of each cocktail drug used in our cocktail were extracted.

#### 2.1.3. Statistical Analysis and Phenotypic Indices Calculation

Metabolite/parent probe drug plasma concentration, metabolic ratios (MR), and area under the curve (AUC) found in the literature were used to calculate 95%,95% tolerance intervals for MR and f-AUC_0–6h_ in order to define “normal” intervals. Human volunteer subjects who were poor or rapid metabolizers based on genetic tools were excluded from our analysis when this information was specified.

A *p*,1–α tolerance interval is an interval that contains *p* % of the population with an 1–α confidence. Here, tolerance intervals were obtained using the formula
[*m* − *k s*; *m* + *k s*]
where *m* is the arithmetic mean of the sample, *s* its standard of deviation, and *k* a factor depending on the sample size, the wanted coverage *p*, and the confidence level, 1–α. This factor was obtained exactly using the tolerance package for R [23]. This formula assumes a Gaussian distribution; hence, it was used on the logarithm of AUC and MR, then exponentiated. Values for *m* and *s* were obtained from the literature, either directly or based on confidence intervals. When several articles gave values for *m* and *s*, values were averaged, weighted by sample sizes. For CYP, tolerance intervals are defined for the AUC_0–6h_ MR and for the single MR point selected in the sensitivity study for the limited sampling strategy. For P-gp, tolerance intervals are defined for AUC_0–6h_ fexofenadine alone and with molecules present in the Geneva cocktail [24].

### 2.2. Part 2: Comparison with Phenotypic Assessment in Patients

#### 2.2.1. Subjects

Patients were included in a prospective study from December 2016 to October 2020 and treated with painkillers in the “Centre d’Évaluation et de Traitement de la Douleur” (Cochin Hospital, Paris, France) or psychotropic drugs in the “Département de Psychiatrie” (Lariboisière—Fernand Widal Hospital, Paris, France) for pain or psychiatric disorders. They all experienced therapeutic issues such as unusual and severe adverse effects, partial or complete lack of efficacy of more than two prescribed drugs with the same indication, abnormal blood drug concentrations above or below normal plasma therapeutic ranges with conventional doses, and possible drug–drug interaction (DDI) due to new co-medicated drugs (see the full list of drugs prescribed to the patients and the list of the possible adverse drug reactions in Appendix A Table A1 and Table A2). Some drug metabolism pathways were explored, based on the clinical situation and the suspicion of one or more drug metabolism abnormalities. This clinical study was approved by the French Ethics Committee “Comité de Protection des Personnes Sud Méditerranée” and numbered ID-RCB-2017-A00685-48. Patients were included after giving their written informed consent. The patients and the clinical outcomes of the phenotypic assessments with psychotropics have been described in a previous study [25].

#### 2.2.2. Study Design

Overnight-fasted subjects did not take caffeine-containing products (coffee, tea, chocolate, energy drinks) for at least 24 h before the study session. Medications were reported as inhibitors or inducers of CYP and P-gp (based on the tool developed by the Division of Clinical Pharmacology and Toxicology, Geneva University Hospitals [26]). Patients who were not already hospitalized were admitted to the hospital in the morning, were given the complete phenotypic cocktail (caffeine 50 mg, bupropion 150 mg, flurbiprofen 50 mg, omeprazole 10 mg, dextromethorphan 10 mg, midazolam 1 mg, fexofenadine 120 mg), and were eventually only given some of the probe drugs that the cocktail contained depending on the metabolic pathways of interest for a given patient. Patients with morning drug intake had to delay the intake until after the first blood collection. Dosages of the drugs used in the cocktail were adapted from the Geneva phenotyping cocktail [21] depending on the availability of drugs at the French hospital pharmacy. Blood samples were collected in lithium heparinate tubes and taken at 2 h, 3 h, and/or 6 h after oral administration of cocktail probe drugs with 250 mL of water. The marketed drugs used were caffeine (CAFEINE CITRATE COOPER^©^ 50 mg/2 mL injectable/oral solution), bupropion (ZYBAN^©^ 150 mg film-coated extended-release tablet), flurbiprofen (CEBUTID^©^ 50 mg coated tablet), dextromethorphan (DRILL^©^ 5 mg/mL syrup, 125 mL or TUSSIDANE^©^ 1.5 mg/mL, 125 mL), omeprazole (OMEPRAZOLE BIOGARAN^©^ 10 mg gastro-resistant capsule), midazolam (MIDAZOLAM PANPHARMA^©^ or MYLAN^©^ 5 mg/5 mL injectable solution), fexofenadine (FEXOFENADINE Zentiva^©^ 120 mg film-coated tablet).

In our study, we decided to use blood rather than urinary samples. As a less invasive method than plasma tests, urinary MR has been widely used in the past and in other cocktails. However, it suffers from a high variability because of several parameters. First, urinary clearance is sensitive to phenomena such as pH changes or renal insufficiency [21]. Moreover, none of the probes and metabolites are eliminated to the same extent in urines (Appendix A Table A3). In the case of dextromethorphan, for instance, the urinary MR is therefore weakly correlated with oral clearance [27]. Because at least one drug had to be evaluated through plasma measurement, we decided to proceed with blood tests for all drugs.

Categorical (gender, smokers, and pain or psychiatric disorders) and continuous (liver and kidney function) variables were collected and described using frequency tables (N, %) and median (range), respectively.

#### 2.2.3. Analytical Methods

The analytical method was adapted from the one published by Bosilkovska et al. [22]. The cocktail probe drugs and their CYP-specific metabolites were quantified in plasma using an in-house validated liquid chromatography–tandem mass spectrometry method (HPLC-MS/MS). Plasma samples were prepared in Eppendorf containing deuterated internal standards. Protein precipitation was performed by adding acetonitrile (ACN), followed by vortex mixing. Samples were centrifuged, and supernatants were transferred to HPLC vials. Analysis was performed using a LC-MS/MS system consisting of a TSQ Quantum Ultra Triple Quadrupole Mass Spectrometer (ThermoFisher, Waltham, MA, USA). Separation was achieved on an Accucore^®^ C18 column (50 × 2.1 mm, 2.6 µm, ThermoFisher) using a mobile phase composed of water containing 0.1% formic acid (A) and acetonitrile containing 0.1% formic acid (B) in gradient elution mode at a flow rate of 0.5 mL/min. Internal standards, MRM transitions (*m*/*z*), collision energies, and polarity mode as well as detection limits and calibration curves are presented in Appendix A Table A4. The lower limit of quantification (LLOQ) was the lowest concentration with an intra- and inter-day coefficient of variation < 20% and intra- and inter-day accuracy within 20% of the nominal value. Accuracies were calculated as percentage deviation of measured concentration from the theoretical original value and did not exceed 15% for QC samples or 20% for the LLOQ. Precisions were evaluated by determining the relative standard deviation (RSD) of each analyte that did not exceed 15% and 20% for QC and LLOQ, respectively.

#### 2.2.4. Pharmacokinetic Analysis

P-gp activity was evaluated by determining fexofenadine AUC from 0 to 6 h after oral administration (f-AUC_0–6h_). CYP activities were assessed by measuring the CYP-specific metabolite/parent drug probe plasma concentrations, yielding single-point metabolic ratios at 2 h, 3 h, and 6 h after cocktail oral administration or AUC_0–6h_ MR as follows (Appendix A Figure A1):-CYP1A2—paraxanthine/caffeine MR (par/caf).-CYP2B6—4-hydroxybupropion/bupropion MR (OH-bup/bup).-CYP2C19—5-hydroxyomeprazole/omeprazole MR (OH-opz/opz).-CYP2C9—4-hydroxyflurbiprofen/flurbiprofen MR (OH-flb/flb).-CYP2D6—dextrorphan/dextromethorphan MR (dor/dem).-CYP3A—1-hydroxymidazolam/midazolam MR (OH-mdz/mdz).

AUC ratios were determined as the ratio between the AUC of the metabolite and the AUC of the administered parent probe drug. AUC_0–6h_ of metabolites and parent drug plasma concentrations were estimated by non-compartmental analysis using the linear up and down trapezoidal method, only for patients who had three sampling times (T2h, T3h, T6h). Phenotyping indices (MR for CYP and f-AUC_0–6h_ for P-gp) are presented as geometric mean and coefficient of variation.

#### 2.2.5. Sensitivity Analysis

To select a single MR sampling time, the influence of each individual MR on the AUC ratio was derived based on the trapezoidal approximation of the AUC. Briefly, the AUC A_X_ for a given molecule X, of concentration *C*_X_(*t*) at time *t*, is approximated by $\frac{3}{2}$*C*_X_(2) + 2*C*_X_(3) + $\frac{3}{2}$*C*_X_(6), assuming sampling at *t* = 2 h, 3 h, and 6 h, and that at *t* = 0, C_X_(*t*) = 0. Hence, the ratio of the AUC for metabolite (X = M) and parent (X = P) is given by *R*_AUC_ = $\frac{3R\left(2\right)C_{P}\left(2\right)+4R\left(3\right)C_{P}\left(3\right)+3R\left(6\right)C_{P}\left(6\right)}{3C_{P}\left(2\right)+4C_{P}\left(3\right)+3C_{P}\left(6\right)}$, where *R*(*t*) = $\frac{C_{M}\left(t\right)}{C_{P}\left(t\right)}$ is the metabolite ratio at time *t*. The effect of each metabolite ratio is then described by the partial derivative, $\frac{\partial R_{\mathrm{AUC}}}{\partial R\left(t\right)}=\frac{x\left(t\right)C_{P}\left(t\right)}{3C_{P}\left(2\right)+4C_{P}\left(3\right)+3C_{P}\left(6\right)}$, with *x*(*t*) = 3 for *t* = 2 h and *t* = 6 h, and *x*(*t*) = 4 for *t* = 3 h. The individual ratio that will most inform the AUC ratio is then the one with the maximal derivative (since it is the one for which a unit change will lead to the highest change in the AUC ratio). Since the denominator is the same for all partial derivatives, the comparison between 3*C*_P_(2), 4*C*_P_(3), and 3*C*_P_(6) yields the answer. This value was computed for each patient, and the most informative time was selected for each patient. The time selected was the one selected for most patients (indicated by “SA”).

#### 2.2.6. Correlation between Single-Point and AUC_0–6h_ MRs

Spearman’s correlation coefficients (r_s_) between single-point and AUC_0–6h_ MRs were calculated for each sampling time. The correlation coefficient was considered significant for *p* < 0.05. The time with the highest Spearman correlation coefficient is indicated by “Sp”.

#### 2.2.7. Comparison to Phenotypic Indices from the Literature

Patients’ MR and f-AUC_0–6h_ distributions were compared with 95%,95% tolerance intervals from the literature analysis. Each patient was classified as “low” (if the value was below the tolerance interval’s lower limit), “normal” (if the value was within the limits of the tolerance interval) or “high” (if the value was higher than the tolerance interval’s upper limit). For CYP, the classification defined using the single-point MR was compared through a concordance study to the classification defined using the AUC_0–6h_ MR. Both classifications obtained by sensitivity analysis and those obtained with the highest Spearman correlation coefficient were compared. The class divergence between single MR and AUC_0–6h_ MR classification (% of mispredicted AUC class) was used to obtain the probability of agreement for each time.

## 3. Results

### 3.1. Part 1: Determination of Phenotypic Indices in Human Volunteers Based on the Literature Analysis

#### 3.1.1. Studies Included

We found 27 studies assessing one or several probe drugs of the Geneva cocktail. Among these studies, 13 assessed CYP1A2 with caffeine (*n* = 231 individuals); 2 assessed CYP2B6 with bupropion (*n* = 40 individuals); 2 assessed CYP2C9 with flurbiprofen (*n* = 40 individuals); 9 assessed CYP2C19 with omeprazole (*n* = 144 individuals); 6 assessed CYP2D6 with dextromethorphan (*n* = 99 individuals); 7 assessed CYP3A with midazolam (*n* = 106); and 9 assessed P-gp with fexofenadine (*n* = 141 individuals). Full drug-by-drug study details can be found in Appendix A Table A5, including study references.

No interaction between molecules of different reviews’ cocktails was found. MRs remained similar regardless of the different doses of probe drugs used in cocktails, except for omeprazole, for which the PK was not linear at high doses when administered in multiple administrations. However, the omeprazole PK remained linear following single or repeated daily doses up to 40 mg. After single or repeated administration, fexofenadine PK remained linear up to 120 mg twice daily. Thus, f-AUC_0–6h_ obtained for various fexofenadine oral doses lower than 120 mg was normalized to a single 120 mg dose as used in our cocktail.

#### 3.1.2. 95%/95%, Tolerance Intervals of Metabolic Indices

The PK data reviewed in the literature were used to establish 95%,95% tolerance intervals of phenotyping indices. These tolerance intervals allowed for the determination of “normal” CYP450 and P-gp activities and are presented in Table 1. For fexofenadine, data in the literature showed differences depending on whether it was administered alone or in the Geneva cocktail. Two 95%,95% tolerance intervals were then defined based on whether the phenotyping of P-gp was assessed using fexofenadine alone or in the Geneva cocktail.

### 3.2. Part 2: Comparison to Phenotypic Assessment in Patients

#### 3.2.1. Subjects

A total of 46 patients (32 women, 14 men) aged 21 to 85 years old were included in the study (Table 2). Twenty patients received all the phenotyping probe drugs, while twenty-six patients received only some of them. Most patients (89%, *n* = 41) had mood disorders. One-fourth were smokers (*n* = 11), and 40% were obese (*n* = 18, BMI ≥ 30 kg/m^2^). Sixteen patients were not taking any treatment considered to induce or inhibit CYP and P-gp activities (according to the classification in [28]).

#### 3.2.2. Pharmacokinetics of Probe Cocktail Drugs and Their Specific CYP-Mediated Metabolites

Pharmacokinetic profiles of cocktail probe drugs and their specific CYP-mediated metabolites are presented in Appendix A Figure A2. Profiles show a strong inter-individual variability in CYP MR and f-AUC_0–6h_ (CV > 60%; Table 3). Results of the selection of single-time MR are shown in Table 3, along with the number of patients for whom the sensitivity coefficient was maximal (n_S_), Spearman’s correlation coefficients (r_s_) between MR AUC_0–6h_, and each single sampling time’s MR. The best time to predict AUC according to the sensitivity analysis was T2h for CYP3A and T3h for the other CYPs. The highest correlation coefficient was at T2h for CYP2B6 and CYP2C19, T3h for CYP1A2, CYP2D6, and CYP3A, and T6h for CYP2C9.

#### 3.2.3. Comparison to Phenotypic Indices from the Literature Analysis

The distribution of the phenotypic indices of patients included in the present study is presented in Figure 1, along with the 95%,95% tolerance intervals determined in healthy volunteers with genotypes corresponding to a “normal” activity. The distribution of patients’ AUC_0–6h_ MR was wider than those expected from the 95%,95% tolerance intervals for CYP2B6, 2C9, 2C19, and 2D6. For selected single-point MRs (the best times to predict AUC according to the sensitivity analysis or the Spearman correlation coefficient), this was also true for CYP1A2 and CYP3A. The classification of patients according to these 95%,95% tolerance intervals is presented in Appendix A Table A6, and the patients’ concordance between the classifications defined with AUC_0–6h_ MRs and those defined with single-point MRs is given in Table 4. Of note, the concordance probability was always higher for the MR selected by the sensitivity analysis, even if the limited sample size does not allow for showing any significant difference.

## 4. Discussion

In the present study, we reviewed and investigated the distribution of P-gp and CYP activities in human volunteers and in patients, respectively, using cocktail probe substrates similar to those used in the Geneva phenotyping cocktail. In 2014, the pioneering work of Bosilkovska et al. assessed CYP1A2/2B6/2C9/3C19/2D6/3A and P-gp activities in ten human volunteers [21]. Phenotyping ranges for “normal”, decreased, and increased CYP and P-gp activities were then proposed using 95% confidence or fluctuation intervals based on means and SD of metabolic ratios for CYP (AUC_metabolite_/AUC_parent drug_) or AUC of fexofenadine for P-gp. Here, we intended to extend this approach to larger population studies in order to determine more precisely the distribution of CYP and P-gp activities in human volunteers and to establish “normal” activity ranges with a high number of subjects. These phenotypic ranges will be useful to predict a metabolic phenotype status in patients at an individual level. We therefore conducted an extensive review of the literature to gather data from 27 clinical studies reporting PK parameters of parent drugs and their CYP-specific metabolites used alone or in phenotyping cocktails. While confidence intervals are usually calculated to provide the bounds of a single-valued population parameter (as the mean), a tolerance interval bounds the range of values that includes a specific proportion of the individual values in a population. We thus determined the 95%,95% tolerance intervals of each of the phenotyping indices in healthy volunteers, meaning that these intervals’ bounds for each MR or f-AUC_2,3,6_ would include 95% of individual values of the entire population with a probability of 95%. The 95%,95% tolerance intervals determined in this study were wider than the “normal” phenotypic ranges determined previously with confidence intervals for all CYP [21], which is as expected, given that they account for inter-individual variability in addition to the experimental precision. A limitation of our approach is that it did not take into account the differences in these studies in terms of study design and heterogeneity of subjects. Even if we decreased this bias as much as possible by including only adult healthy volunteers, receiving single-dose administration, without co-medication, several variability factors were not taken into account such as age, weight, and genotypes. Recently, Lorenzini et al. assessed the phenotyping CYP activity in more than 500 patients with various clinical situations. The percentage of patients with “normal” CYP activities ranged from 57% to 84%, depending on the CYP of interest [29], in agreement with our approach. The important inter-individual variability in phenotyping indices in our study can be explained by the inclusion of patients instead of healthy volunteers without any genetic variants affecting the activity of some CYP (CYP2D6, CYP2C19) and the presence of co-medications containing either inhibitors or inducers of CYP and P-gp. This was clearly shown in ambulatory patients stably treated with antidepressants [30].

When we compared the metabolic ratio in patients to tolerance intervals, ratios of patients for CYP1A2 and CYP3A based on AUC or single MR at T3h were well captured by tolerance intervals, whereas those of CYP2C9, CYP2C19 and CYP2D6 were clearly out of the tolerance intervals, suggesting that genetic variants and/or co-medications with inhibitors of these CYP might explain such results. Additionally, f-AUC_0–6h_ of fexofenadine in patients was mostly higher than the upper limit of the tolerance intervals, suggesting that P-gp intestinal activity was reduced in half of our patients. Despite the fact that no relationship between demographic factors and co-medications affecting CYP or P-gp activities could explain the phenotypic indices, the low number of patients included in this study is probably a limitation for reaching this goal. The absence of genotyping tools did not allow us to understand the mechanism of poor drug metabolism observed for some patients in this study, but the objective of this study was not to correlate genotypes with phenotypes. Recently, relationships between genotyping-based phenotypes and measured phenotypes were determined in patients treated with antidepressants [31], showing that for all CYP tested, an important rate of phenoconversion occurred in between 33% and 65% of the patients. Regardless of the reason why phenotyping indices for some patients were out of the “normal” activity range, as determined in healthy volunteers, our study allows us to determine patients with induced or reduced CYP and P-gp activities. The use of such ranges for phenotyping indices in the clinical setting might thus be applied to patients experiencing no effect or adverse effects of newly introduced medications, in order to search for a therapeutic issue based on abnormal drug metabolism that would contribute to decreasing or increasing the formation of an inactive or active metabolite. The phenotypic cocktail in the clinical practice can guide the choice of different molecules to prevent toxicity or therapeutic failure. This approach would be even more interesting for patients whose phenotyping indices are clearly out of the bounds of our 95%,95% tolerance intervals. The interpretation of patient indices at the limit of the intervals remains more complex and must be conducted with clinical data. Tolerance intervals determined from a high number of clinical PK studies also allow for the classification of patients in further clinical studies based on their CYP and P-gp activities. 

Phenotyping has the advantage of taking co-medications, epigenetics, environment, and genetics into account. A genetic mutation does not necessarily have phenotypic consequences, and the influence of a drug on a metabolic pathway cannot be evaluated through genetics, hence our interest in this phenotyping approach as previously demonstrated [31]. This phenotyping assay should be conducted as smoothly as possible for clinicians and patients, and multiple sampling in one complete day should be avoided. This is why we looked at the possibility of using only one blood sample instead of three to accurately assess CYP and P-gp phenotyping indices. A very limited sampling strategy with one sampling time seems sufficient in patients, as was already reported in healthy volunteers. Patient results are mostly concordant between single-point MRs and MR AUC_0–6h_ for CYP-probe or f-AUC_2,3,6_ and f-AUC_2,3_ for P-gp probe. Our correlation and concordance analysis revealed that the best sampling time was at T3h for predicting phenotyping indices based on AUC_2,3,6_ for CYP1A2, CYP2C19, CYP2D6, and CYP3A, as was also previously established in healthy volunteers. For CYP2B6, almost half of patients’ results were mispredicted when considering T2h or T3h as a surrogate for AUC_2,3,6_. Single MR of CYP2B6 at T3h was higher than the “normal range” for half of the patients, whereas it was not observed when considering AUC_2,3,6_, suggesting that the kinetics of bupropion and its CYP2B6-mediated hydroxyl-bupropion would differ in volunteers and in patients. For CYP1A2, missing data in our correlational study (*n* = 11) prevented us from confirming T2h as the best sampling time (best m_S_ at time 3 h and ρ_s_ 0.56 *p*-value 0.075 at T2h). 

The phenotypic assessment of metabolic pathways has demonstrated its clinical value [25,32,33]. Although it is challenging to make a direct comparison between phenotyping and genotyping regarding clinical outcomes, the high phenoconversion rate [34] leads us to assert that the phenotypic evaluation of CYP and P-gp activity is a more effective tool than genotyping for personalized medicine [35]. An extensive literature review allowed us to establish tolerance ranges in order to define phenotypic indices for the drugs used in the Geneva cocktail in clinical practice, and it revealed a greater variability of phenotypic indices in our patients. Interpretation of phenotypic indices in patients needs to be combined with clinical data and information on respective values for biomarkers of liver function/dysfunction that could have important consequences in drug biotransformation.

## Figures and Tables

**Figure 1 pharmaceutics-15-00979-f001:**
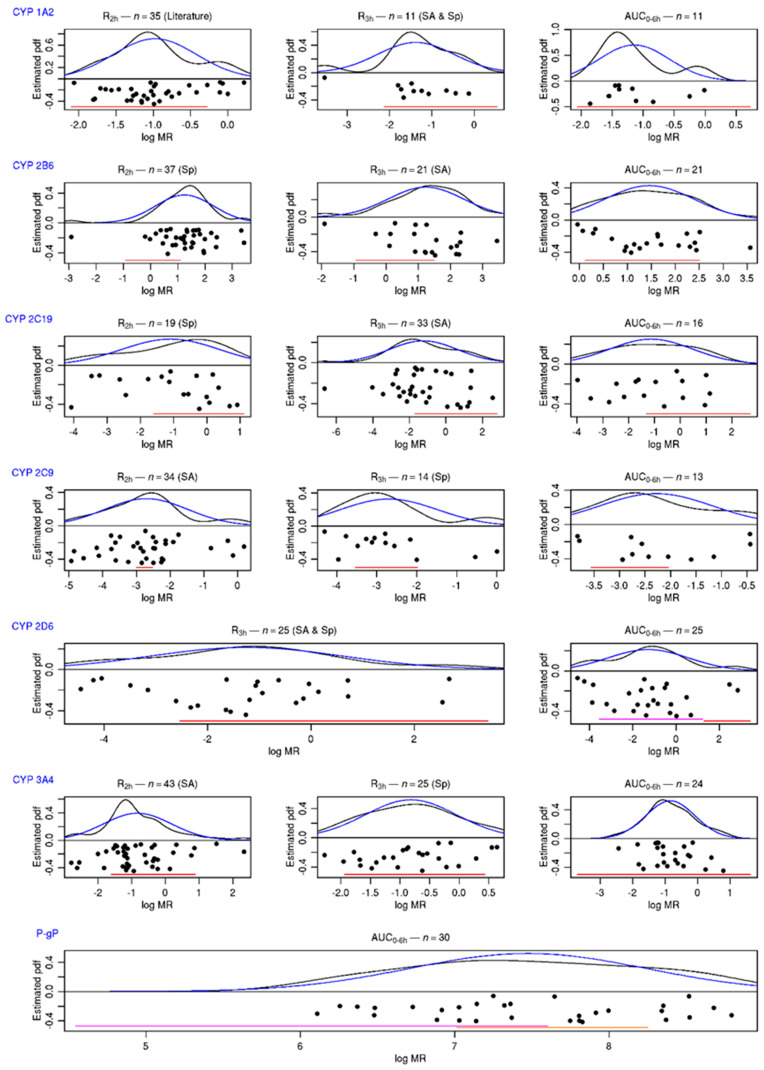
Distribution of phenotyping indices for “normal” activities obtained from the literature analysis. Distributions of patients’ MRs and f-AUC_0–6h_ (black dots) are presented in logarithmic scale. In black: the estimation of distribution function by the kernel estimator. In blue: the probability density of the estimated Gaussian. In red: the 95%,95% tolerance interval obtained from the literature (for CYP2D6, IM in purple and EM in red; for P-gp, AUC_0–6h_ for fexofenadine in the Geneva cocktail in red and AUC0–6h for fexofenadine alone in orange; for CYP 2B6 and CYP 2C9 t_2h_, the interval from Geneva studies in red). SA: best time to predict AUC MR, according to sensitivity analysis. Sp: highest Spearman correlation coefficient with AUC MR. Literature: time to predict AUC MR in literature.

**Table 1 pharmaceutics-15-00979-t001:** Phenotyping indices predicted intervals for “normal” activities obtained from the literature analysis.

CYP450 (Cocktail probe)	**MRs**	**Decreased Activity**	**Normal Activity**		**Increased Activity**
CYP1A2 (caffeine)	AUC_0–6h_	<0.13	0.13–2.07		>2.07
	t2h	<0.12	0.12–0.76		>0.76
	t3h	<0.12	0.12–1.72		>1.72
CYP2B6 (bupropion)	AUC_0–6h_	<1.13	1.13–12.22		>12.22
	t3h	<0.39	0.39–4.45		>4.45
CYP2C19 (omeprazole)	AUC_0–6h_	<0.27	0.27–15.09		>15.09
	t2h	<0.20	0.20–3.06		>3.06
	t3h	<0.18	0.18–16.49		>16.49
CYP2C9 (flurbiprofen)	AUC_0–6h_	<0.03	0.03–0.13		>0.13
	t3h	<0.03	0.03–0.14		>0.14
CYP2D6 (dextromethorphan)	AUC_0–6h_	<0.03	IM 0.03–3.4	EM 3.4–31.2	>31.2
	t3h	<0.08	0.08–30.6		>30.6
CYP3A (midazolam)	AUC_0–6h_	<0.02	0.02–5.10		>5.10
	t2h	<0.20	0.20–2.42		>2.42
	t3h	<0.14	0.14–1.54		>1.54
Transporter (molecule)	**AUC** **(µg/L*h)**	**Decreased Activity**	**Normal Activity**		**Increased Activity**
P-gp (120 mg) (fexofenadine in Geneva cocktail)	AUC_0–6h_	>1538	167–1538		<167
P-gp (120 mg) (fexofenadine alone)	AUC_0–6h_	>3486	1229–3486		<1229

EM: extensive metabolizers; IM: intermediate metabolizers.

**Table 2 pharmaceutics-15-00979-t002:** Sociodemographic and clinical characteristics (*n* = 46 patients).

	Median	Range
Age (years)	49	21–85
BMI (kg/m^2^)	27.4	17.9–41.6
AST (UI/mL)	24	15–92
ALT (UI/mL)	22	5–160
GGT (UI/mL)	35	10–211
PAL (UI/mL)	73	38–113
Albumin (mg/L)	40	17–51
Protein (mg/L)	67	53–85
CKD-Epi (mL/min/1.73 m^2^)	85	52–130
	N	%
Female	32	70
Smoker	11	24
Psychiatric disorders	41	89
Pain disorders	5	11

**Table 3 pharmaceutics-15-00979-t003:** Phenotyping indices for CYP and P-gp probe drugs and sensitivity analysis and correlation between single-time and AUC_0–6h_ MRs.

CYP or Transporter	Number of Subjects	Sampling Time	Phenotyping Indices		Sensitivity Analysis			Correlation between Single-Time and AUC_0–6h_ MRs	
			Geo Mean ^a^	CV (%) ^b^	m_SA_	CV (%)	n_SA_	r_S_	p_S_
CYP1A2	34 11 11 11	T2h T3h T6h AUC_0–6h_	0.37 0.25 0.40 0.32	61 74 77 70	1442 **4686** 3030	88 **78** 63	1 (9%) **8 (73%)** 2 (18%)	0.56 **0.91** 0.72	0.075 **<0.001 *** 0.017 *
CYP2B6	36 21 21 21	T2h T3h T6h AUC_0–6h_	3.39 3.44 7.90 4.30	120 116 117 114	201 **335** 148	108 **75** 85	4 (19%) **17 (81%)** 0 (0%)	**0.98** 0.94 0.95	**<0.001 *** <0.001 * <0.001 *
CYP2C19	19 32 23 16	T2h T3h T6h AUC_0–6h_	0.34 0.29 0.45 0.32	101 203 167 126	357 **720** 285	235 **200** 197	7 (43.7%) **7 (43.7%)** 4 (25%)	**0.95** 0.94 0.89	**<0.001 *** <0.001 * <0.001 *
CYP2C9	33 14 13 13	T2h T3h T6h AUC_0–6h_	0.07 0.07 0.15 0.10	174 176 134 120	4206 **10,760** 4806	104 **63** 71	1 (8%) **9 (69%)** 3 (23%)	0.34 0.66 **0.93**	0.255 0.017 * **<0.001 ***
CYP2D6	40 25 25 25	T2h T3h T6h AUC_0–6h_	0.37 0.29 0.23 0.28	275 241 309 256	5 **10** 6	126 **98** 111	2 (8%) **23 (92%)** 0 (0%)	0.97 **0.99** 0.91	<0.001 * **<0.001 *** <0.001 *
CYP3A	42 25 25 24	T2h T3h T6h AUC_0–6h_	0.43 0.44 0.54 0.43	197 81 256 86	**7** 5 2	**95** 141 109	**22 (92%)** 2 (8%) 0 (0%)	0.94 **0.98** 0.90	<0.001 * **<0.001 *** <0.001 *
P-gp	30	f-AUC_0–6_	1760	75					

MR: metabolic ratio. f-AUC_0–6_: AUC of fexofenadine plasma concentrations from 0 to 6 h after intake; ^a^ geometric means; ^b^ coefficient of variation. n: number of patients usable for the analysis. m_SA_: average sensitivity coefficient. CV: m_SA_ coefficient of variation. n_SA_: number of patients with the highest sensitivity coefficient. r_S_: Spearman’s correlation coefficient. p_S_: p of Spearman’s correlation test. In bold, the best sensitivity and/or Spearman’s coefficients. * statistically significant

**Table 4 pharmaceutics-15-00979-t004:** Concordance table between the classification defined using AUC_0–6h_ MR and the classification defined using single-point MR.

CYP450	Single-Point MR	n	Agreement Probability	Number (%) of Mispredicted AUC Class	Best Time to Predict AUC according to:
CYP1A2	T2h	11	0.82 [0.48; 0.98]	2 (18%)	
	T3h	11	0.91 [0.59; 0.99]	1 (9%)	SA; Sp
CYP2B6	T2h	21	0.52 [0.30; 0.74]	10 (47%)	Sp
	T3h	21	0.62 [0.38; 0.82]	8 (38%)	SA
CYP2C19	T2h	16	0.81 [0.54; 0.96]	3 (18%)	Sp
	T3h	16	1	0 (0%)	SA
CYP2C9	T3h	13	0.77 [0.46; 0.95]	3 (23%)	SA
	T6h	13	0.54 [0.25; 0.81]	6 (46%)	Sp
CYP2D6	T3h	25	0.92 [0.70; 0.98]	2 (8%)	SA, Sp
CYP3A	T2h	24	0.88 [0.68; 0.97]	3 (12%)	SA
	T3h	24	0.83 [0.63; 0.95]	4 (16%)	Sp

Probabilities are given with their 95% confidence interval. SA: best time to predict AUC MR, according to sensitivity analysis. Sp: highest Spearman correlation coefficient with AUC MR. Literature: time to predict AUC MR in literature.

## Data Availability

Not applicable.

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
