# Peer review of "Phenotyping Indices of CYP450 and P-Glycoprotein in Human Volunteers and in Patients Treated with Painkillers or Psychotropic Drugs"

_pharmaceutics, 2023, doi:10.3390/pharmaceutics15030979_

Round 1

Reviewer 1 Report (Previous Reviewer 3)

This version is significantly improved. Still, I would like to see some comments on the cost-effectiveness of this approach, especially in the context of genotyping vs. phenotyping. 

Conclusion, should gave some statement regarding the benefits and place of this approach in clinical practice. Since, it looks complicated in comparison to genotyping. Maybe, I am wrong. Surely, it is more precise approach than genotyping, including gene variability within.

Author Response

Reviewer 2 Report (Previous Reviewer 2)

I think the paper looks much better now. I have noticed language pitfalls; therefore, its important to get the manuscript carefully reviewed before publications.   

Author Response

This manuscript is a resubmission of an earlier submission. The following is a list of the peer review reports and author responses from that submission.

Round 1

Reviewer 1 Report

This is an interesting and well written article that highlights the importance of drugs’ combinations probes allowing the in vivo assessment of multiple pathways of drug metabolism, as a phenotyping "cocktail" containing in this study:  caffeine, bupropion, flurbiprofen, omeprazole, dextromethorphan, midazolam and fexofenadine. The following points may potentially be addressed and/or modified:

1-         In this work, the measurements of parent drugs and respective metabolite concentrations in urine, were not undertaken for assessment of phenotyping. Thus, the authors may want to discuss in higher detail on the advantages and disadvantages and/or limitations of the sampling strategy used in the experimental study (only blood samples were collected and taken at 2 h, 3 h up to 6 h after oral administration). A short table gathering the consensual major kinetic parameters for this group of probe drugs would be of help.

2-         Authors mention absence of genotyping tools as a limitation of the study. A short discussion on potential variant genotypes observed in patients with implications for respective conclusions, could be briefly added.

3-         It would be helpful to the reader, to include a scheme, at least as supplementary figure, with the main metabolic pathways and excretion reactions for the studied drugs, clarifying: i) the chemical structures of the molecules; ii) the respective metabolyzing enzymes/transporters; iii) the relative quantitative contribution of the assessed CYP activity on overall drug metabolism.

4- The details of patients’ medications (beyond general designation as pain killers or psychotropic drugs), referred on table 1 as “drugs having inhibitor or inducer properties on CYP and P-gp”, should be made clearer.

5- As conclusion, authors state that the “Interpretation of phenotypic indices in patients needs to be combined with clinical data” (last sentence). Therefore, I strongly suggest the inclusion of further clinical information concerning this target group of patients, including for instance, information on respective values for biomarkers of liver function/dysfunction that could have important consequences in drug biotransformation.

Reviewer 2 Report

Thank you for giving me the the chance to review this interesting paper. The authors have done extensive work trying to explore the phenotypic activity of the main CYP involved in drug 78 metabolism (CYP1A2, CYP2B6, CYP2C9, CYP2C19, CYP2D6, CYP3A) and the P-gp in 79 patients having therapeutic issues with painkillers or psychotropic drugs. In my opinion this work should not be considered for publication in its current status for the following reasons:

  • There was a mismatch between the study main objective and their conclusion. 
  • The novelty was absent from this work as all findings were already known and well studied previously. 
  • The authors tried to identified a general tolerance ranges for the phenotypic activity of the included CYP enzymes extracting from previous studies without taken into account the differences in these studies in term of study designs, heterogeneity, …. etc. 
  • In term of paper writing, I have found that the authors included so much details that I found distracting for readers and at some points taking me away from the main study objective. 

Reviewer 3 Report

Dear Authors,

It is good an quality written paper with an interesting topic to deal with. It also provides some insights for the future studies. Still, I have some concerns:

1. The aim of the study should be clearly stated (separately or at the end of introduction part)

2. Which painkillers and psychotropic drugs were used and patients experience adverse effects, etc. as you have stated in Human subjects section of Methodology. Are those patients the same, who were given the drug cocktails. Should be stated clearly. 

3. Abstract should be more informative due to some readers will only read abstract not the whole paper and what I conclude that you have offer a strategy to estimate phenotypic indices instead doing genotyping. Did I understand correctly? it is important to be emphasized in the introduction due to almost none of the patients have one drug in their treatment.

4. As I am said before you should write more detailed why is this important, clinical relevance, from the aspect of the real patients.